# Hierarchical Context Merging: Better Long Context Understanding for Pre-trained LLMs

**Woomin Song**[1,*] **Seunghyuk Oh**[1,*] **Sangwoo Mo**[2] **Jaehyung Kim**[3,†]
**Sukmin Yun**[4,‡] **Jung-Woo Ha**[5] **Jinwoo Shin**[1]
[1]KAIST [2]University of Michigan [3]Carnegie Mellon University
[4]Hanyang University ERICA [5]NAVER

## Abstract

Large language models (LLMs) have shown remarkable performance in various natural language processing tasks. However, a primary constraint they face is the context limit, i.e., the maximum number of tokens they can process. Previous works have explored architectural changes and modifications in positional encoding to relax the constraint, but they often require expensive training or do not address the computational demands of self-attention. In this paper, we present *Hierarchical cOntext MERging (HOMER)*, a new training-free scheme designed to overcome the limitations. HOMER uses a divide-and-conquer algorithm, dividing long inputs into manageable chunks. Each chunk is then processed collectively, employing a hierarchical strategy that merges adjacent chunks at progressive transformer layers. A token reduction technique precedes each merging, ensuring memory usage efficiency. We also propose an optimized computational order reducing the memory requirement to logarithmically scale with respect to input length, making it especially favorable for environments with tight memory restrictions. Our experiments demonstrate the proposed method's superior performance and memory efficiency, enabling the broader use of LLMs in contexts requiring extended context. Code is available at https://github.com/alinlab/HOMER.

## 1 Introduction

In recent years, large language models (LLMs) have performed exceptionally in various natural language processing tasks (OpenAI, 2023; Touvron et al., 2023). Using this capability, multiple emerging applications are using LLMs as a central component. However, LLMs have a fundamental constraint in their context limit, which means the maximum number of input tokens they can process. The ability to handle long contexts is important for real-world applications: chatbots might need to interpret extensive chat histories, while the user could task code comprehension models to process extensive codebases.

A significant challenge in overcoming the context limit is addressing the quadratic computational burden of the self-attention mechanism. Prior works have attempted to reduce the computational cost by altering the model architecture, such as introducing sparse attention (Child et al., 2019; Beltagy et al., 2020) or linearized attention (Kitaev et al., 2020; Katharopoulos et al., 2020). Yet, such methods are often not scalable (Tay et al., 2022), and more importantly, they often require extensive model training, making them difficult to use for large-scale models that are prevalent today.

To overcome this issue, recent works have focused on strategies to extend the context limit of pre-trained state-of-the-art LLMs. However, their major focus has been modifying the positional encoding (Chen et al., 2023; Peng et al., 2023), which does not address the quadratic computational cost of self-attention, leaving the efficiency concern unaddressed. Reducing the complexity of pre-trained LLMs remains an important yet underexplored research question.

---

[*]Equal contribution.
[†]Work done in KAIST.
[‡]Work done in Mohamed bin Zayed University of Artificial Intelligence.

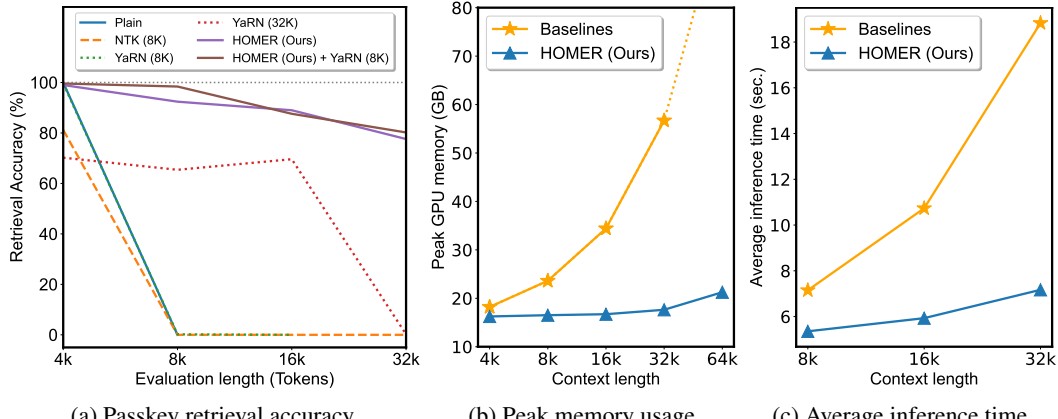

Figure 1: (a) Passkey retrieval accuracy on various context lengths, measured with Llama-2-7b-chat. HOMER maintains reasonable performance for context lengths up to 32K tokens. Detailed comparisons with more baselines are provided in Table 1. (b) The memory requirement for processing long inputs. (c) Average inference time required for generating 100 tokens conditioned on various context lengths. All efficiency measurements are done with a single A100 GPU. The baselines include plain Llama, PI, NTK, and YaRN. Peak memory usage of the baselines at 64k is an estimated value, as they do not fit in a single A100 GPU. Detailed results are provided in Table 5 and Appendix E.

In this paper, we introduce HOMER (**H**ierarchical c**O**ntext **MER**ging), a novel technique designed to extend the context limit while ensuring computational efficiency. HOMER employs a divide-and-conquer approach, dividing the long input into manageable chunks. Unlike previous methodologies (Wang et al., 2023; Bertsch et al., 2023), HOMER does not process these chunks independently. Instead, it employs a hierarchical merging strategy, progressively merging adjacent chunks as they are processed along the transformer layers (see Figure 2 for its illustration). To ensure computational efficiency, apply token reduction before each merging stage.

Furthermore, HOMER can be applied to pre-trained LLMs without any further finetuning. This can be beneficial for practical use scenarios where model finetuning is infeasible, such as in environments with limited computing resources. Also, data preparation may present another challenge for finetuning due to the scarcity of coherent texts with tens of thousands of tokens. For instance, specialized text data should be prepared to finetune an instruction-finetuned or chat-finetuned model without severely losing its desired properties.

Through extensive evaluation on downstream tasks and perplexity measurements, we demonstrate that HOMER can effectively extend pre-trained LLMs to handle long inputs beyond their context limits. We first verify the effectiveness of our method on various downstream tasks, including passkey retrieval and question answering. We further demonstrate the fluency of HOMER by measuring perplexity on long documents. Finally, we highlight the computational efficiency of HOMER as presented in Figure 1b and Figure 1c. In all experiments, we illustrate that HOMER can be used with conventional positional encoding scaling techniques (Chen et al., 2023; bloc97, 2023; Peng et al., 2023), and shows improved performance when used on top of these approaches.

In summary, our contributions are as follows:

- We present hierarchical context merging: a memory-efficient context limit extension technique, that can be used with pre-trained LLMs without additional training.

- We assess the effectiveness of HOMER through experiments on long inputs. In passkey retrieval experiments, HOMER shows 80.4% retrieval accuracy for 32k inputs, whereas even the best-performing baseline shows only 22.4% accuracy. HOMER also improves the prediction accuracy on question answering by 3% (32.7% → 35.7%), presenting its capability to perform complex reasoning about the content in the extended context length. In language modeling experiments, HOMER is the only method showing low perplexity on inputs up to 64k tokens, while the baselines exhibit severe performance degradation for inputs over 32k tokens.

- We demonstrate the efficiency of our approach and analyze the source of computational savings. Utilizing an optimized computation order, memory requirement scales logarithmically with respect to the input sequence length, reducing the memory requirement by over 70%.
- We show that our method is compatible with the conventional RoPE-scaling methods in a plug-in manner, and using them together achieves an additional performance gain.

## 2 RELATED WORK

**Long-range transformers.** Classical methods for long-range transformers primarily focus on reducing the quadratic computational cost of self-attention, such as sparse attention (Dai et al., 2019; Child et al., 2019; Rae et al., 2019; Qiu et al., 2019; Beltagy et al., 2020; Zaheer et al., 2020), or linearized attention (Kitaev et al., 2020; Katharopoulos et al., 2020; Wang et al., 2020; Choromanski et al., 2021). However, these approaches fundamentally change the underlying architecture, and it has not been proven to be scalable for large models (Tay et al., 2022).

**Extension of LLM context lengths.** As the context limit of LLMs has become a critical problem, a line of concurrent works emerged, focusing on efficiently extending the context length of LLMs, with most works focusing on Llama (Touvron et al., 2023). Most works focus on scaling the Rotary Position Embedding (RoPE) (Su et al., 2021). Chen et al. (2023) and kaiokendev (2023) concurrently discovered the Position Interpolation method (PI), which involves linearly interpolating the position ids. bloc97 (2023) suggested an NTK-aware scaling method (NTK) which further alters the base of RoPE. Peng et al. (2023) further extended NTK-aware scaling, suggesting another RoPE scaling method, YaRN. Several works additionally alter the attention mechanism by either applying a mask (Han et al., 2023) or setting an upper bound on the distance between tokens (Su, 2023).

While all methods are known to work without further training, we consider PI, NTK, and YaRN as our main baselines as they are directly compatible with Flash Attention 2 (Dao, 2023), easily enabling memory-efficient inference on long inputs. We also emphasize that our work is orthogonal to these work, and can be further applied on top of these methods to further improve performance.

**Divide-and-conquer approaches.** Approaches to overcome the quadratic computation problem in long context modeling while using the same quadratic self-attention mechanism are to divide the long input into multiple chunks, and most methods process the chunks independently. Inspired by Fusion-in-Decoder (Izacard & Grave, 2020), SLED (Ivgi et al., 2023) independently encodes multiple chunks and feeds all of them to the decoder. Similarly, Unlimiformer (Bertsch et al., 2023) introduces a k-NN search on the encoder outputs, reducing the number of visible tokens at inference time. Retrieval-augmented LLMs including Memorizing transformers (Wu et al., 2022) and LongMem (Wang et al., 2023) take a similar approach of individually forwarding each chunk, and retrieve the cached hidden states for further use. Most of these methods, except for Unlimiformer, require method-specific finetuning.

**Token reduction.** Token reduction methods have been widely studied in the field of efficient vision transformers. The key idea of these methods is to progressively reduce the number of tokens in order to reduce computation, resulting in more efficient training and inference. Two main approaches in this direction are either pruning the redundant tokens (Liang et al., 2022) or merging them (Bolya et al., 2022). To the best of our knowledge, this is the first work to apply token reduction to extend the context limit of large language models.

## 3 HIERARCHICAL CONTEXT MERGING

In this section, we illustrate the detailed procedure of our proposed method, **H**ierarchical c**O**ntext **MER**ging (HOMER); a novel and efficient method for extending the context limit of large language models (LLMs). As visualized in Figure 2, HOMER consists of two steps: (i) hierarchical merging of the intermediate hidden states, which we call *context embeddings*, and (ii) further refinement of the lower-layer embeddings by propagative refinement to produce a compact, fixed-length embedding for each layer, which can be seamlessly integrated as a typical kv-cache (Chen, 2022). We first introduce the key idea of hierarchical merging in Section 3.1. Then, we explain propagative refinement in Section 3.2. Finally in Section 3.3, we introduce an optimized computation order to further reduce the memory requirement to scale logarithmically with the input length.

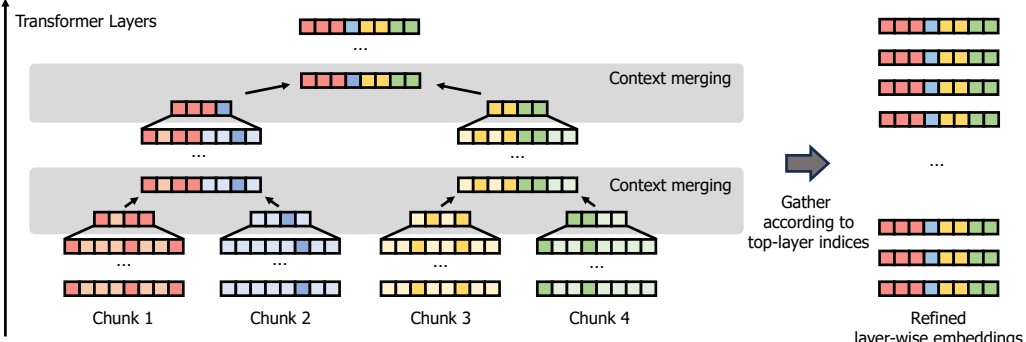

Figure 2: **An overview of the proposed hierarchical context merging.** We first divide a long context into multiple chunks and independently forward them through the early transformer layers. In the intermediate layers, we merge multiple chunks by concatenation, forming a new, *merged chunk*. To keep the chunk length bounded, we apply token reduction on the original chunks to make them shorter, prior to merging. This process is repeated until all chunks are merged into a single chunk. Finally, we further refine the lower-layer embeddings to get a compact fixed-length, layer-wise embedding. The embedding can then be used like a standard kv-cache (Chen, 2022).

## 3.1 HIERARCHICAL MERGING OF CONTEXT EMBEDDINGS

We propose a divide-and-conquer approach to handle the quadratic computation of self-attention more efficiently. We divide the long input into multiple chunks and process the local chunks with the usual self-attention. Although some previous studies have adopted a similar approach (Ivgi et al., 2023; Bertsch et al., 2023), they independently handle each chunk, possibly restricting the richness of the intermediate embeddings as they only have access to local information. In contrast, we progressively merge adjacent chunks as they move through the transformer layers, enabling the chunks to see each other. However, naïvely concatenating the adjacent chunks lengthens the resulting chunk and adds a significant computational burden. Thus we propose to use a token reduction technique to shorten each chunk before merging.

By hierarchically reducing and merging the context embeddings, our method bypasses the quadratic computations required by the self-attention mechanism. This approach not only aims at computational efficiency but also preserves the richness of the context. The detailed process of hierarchical context merging is carried out as follows.

**Division of long context into multiple chunks.** The first step of our method is to divide the long context into uniform chunks. However, simply slicing the input into chunks encounters issues in the network's initial layers where each chunk cannot see each other. This approach restricts most tokens from accessing the starting instructions, harming the resulting embeddings' quality. Moreover, the tokens at the end miss the global context, which is essential for generating subsequent tokens. We address this by attaching the initial and concluding parts of the prompt to every segment (i.e., treating them as shared prefixes and suffixes), ensuring each chunk contains the instruction and the ending tokens.

**Token reduction on individual chunks.** To keep the resulting chunk's length short after merging, we adopt token reduction techniques, which have been widely studied in the field of efficient vision transformers. For vision transformers (Dosovitskiy et al., 2021), dropping the tokens that receive minimal attention from the [CLS] token (i.e. the classification token) is known to be a simple and effective token pruning method (Haurum et al., 2023). Inspired by this, we propose to prune the tokens receiving minimal attention from the final token in each chunk. If the chunks contain affixes, we do not prune the tokens corresponding to the affixes.

In practice, we identified a position bias in simple attention-based pruning where tokens near the end often receive higher attention weights. To rectify this, We incorporate a calibration technique

---

**Algorithm 1** Memory-efficient computation ordering

---

**procedure** HIERARCHICALMERGE(node)
    **if** node is a leaf **then**
        emb_list ← ForwardLayers(node)                ▷ Get layer-wise hidden states
        emb_list ← Refine(emb_list)                   ▷ Propagative refinement
    **else**
        l_embs ← HierarchicalMerge(node.left)         ▷ Recursively merge children
        r_embs ← HierarchicalMerge(node.right)
        emb_merged ← concatenate(l_embs, r_embs)                ▷ Merge chunks
        emb_list ← ForwardLayers(emb_merged)     ▷ Get layer-wise hidden states
        emb_list ← Refine(emb_list)                   ▷ Propagative refinement
    **end if**
    **return** emb_list
**end procedure**

---

inspired by (Zhao et al., 2021). By averaging the attention weights of the last token with respect to the tokens at each position ahead, we derive the *bias logits*. These bias logits are subtracted from the attention logits during the token reduction to refine token pruning. In summary, the final token pruning is performed by pruning a fixed number of tokens according to the significance score $s_{\texttt{sig}}$ defined as follows:

$$s_{\texttt{sig}}^{\texttt{i}} := l_{\texttt{att}}^{\texttt{i}} - l_{\texttt{bias}}^{\texttt{dist(i)}}, \tag{1}$$

where $s_{\texttt{sig}}^{\texttt{i}}$ denotes the significance score of a token at position $\texttt{i}$, $l_{\texttt{att}}^{\texttt{i}}$ denotes the token's attention logit, and $l_{\texttt{bias}}^{\texttt{dist(i)}}$ denotes the bias logit corresponding to the token's distance from the final token, $\texttt{dist(i)}$.

**Merging chunk embeddings.** After shortening, adjacent chunks are concatenated to form a unified chunk. This iterative process of reduction and merging across layers ensures individual chunks converge into a single chunk at the final layers. If the chunks include affixes, direct concatenation might lead to redundancy; we address this by simply averaging the duplicates.

**Handling position ids.** Management of position ids is an important design choice for our approach. While dynamically scaling the position ids through conventional methods like PI, NTK, and YaRN is viable, these techniques tend to underperform with increased scale factors, being less effective for extended contexts. To circumvent this issue, we reuse the same position ids across different chunks. For affixes, we ensure that corresponding tokens in different chunks are assigned the same ids for consistency across the chunks.

## 3.2 PROPAGATIVE REFINEMENT OF LOWER-LAYER EMBEDDINGS

As depicted in Figure 2, the hierarchical context merging produces embeddings characterized by a trapezoidal shape. The higher-layer embeddings are concise, while the lower-layer ones remain extended. To further reduce the computational burden for lower layers, we introduce an additional refinement step after token reduction, called *propagative refinement*.

The process is straightforward: when a token is pruned in the upper layers, the corresponding tokens are also pruned in the lower-layer embeddings. Therefore, the pruning decision of the upper layers propagates back to the lower layers. The synchronized pruning across layers results in shorter, uniform embeddings for each layer. For better understanding, we have added a detailed illustration in Appendix D demonstrating the process step-by-step. The rationale behind this is an intuition that the upper layers have a better ability to identify the important tokens. Thus, we apply pruning in the upper layers and reflect them in the lower layers. After performing hierarchical merging and propagative refinement, we end up with standardized, fixed-length embeddings for every layer.

**Using the refined embeddings for further generation.** Conventional implementation of autoregressive language models often cache the key and value embeddings in order to avoid redundant computation. This technique is commonly known as *kv-caching* (Chen, 2022). As the refined embeddings have the same length for every layer, they can easily be integrated with the kv-cache implementation by simply replacing it for the generation process.

### 3.3 COMPUTATION ORDER OPTIMIZATION FOR MEMORY-LIMITED ENVIRONMENTS

In typical Transformer models, all tokens at a given layer are computed in parallel. Following this paradigm, a direct implementation of HOMER would also process multiple chunks concurrently. While such implementation of HOMER inherently requires linear memory with respect to the input length, we propose a more optimized computation order that allows the memory requirement to scale logarithmically. This efficiency is achieved by strategically reordering the processing steps during the hierarchical merging.

While representing each chunk as a node, the hierarchical context merging process can be conceptualized as a traversal on the binary tree from leaves to the root. By adopting the depth-first search (DFS) algorithm to the computation sequence while executing the propagative refinement, we can achieve a computation cost of a logarithmic scale with respect to the length of the input sequence. For clarity, a pseudo-code representation is provided in Algorithm 1 and Figure 3. A comprehensive proof of the memory requirement can be found in Appendix A. Through this approach, extensive inputs can be processed even in resource-constrained setups.

## 4 EXPERIMENTS

In this section, we demonstrate the effectiveness of the proposed method, HOMER through extensive experiments. Section 4.1 contains the passkey retrieval experiments, originally suggested by Mohtashami & Jaggi (2023). This shows our method's ability to utilize the long context to handle downstream tasks. Section 4.2 contains experiments on question answering. This shows the model's capability to handle more complex and challenging tasks. Section 4.3 demonstrates that HOMER remains fluent, even when conditioned on very long contexts. This is done by measuring perplexity on long documents from PG-19 dataset (Rae et al., 2019). Section 4.4 contains ablation study on the key components that make HOMER effective. Finally in Section 4.5, we analyze the memory efficiency of our method.

**Common setup and baselines.** We select Llama-2 as our base model, as it is the most widely used and the strongest open-source large language model. We use the pretrained models for language modeling experiments, and the chat model for evaluation on downstream tasks, which include passkey retrieval and question answering.

Recent works on positional encoding interpolation have shown their ability to extend Llama's context limit without training. We set Position Interpolation (PI) (kaiokendev, 2023), NTK-aware scaling (bloc97, 2023), and YaRN (Peng et al., 2023) as our main baselines. As these models scale the positional encoding by a constant factor, we define their context limit as the original context limit (4k tokens for Llama-2) multiplied by the scaling factor. In practice, NTK and YaRN are known to be able to process slightly shorter context than the defined context limit (Peng et al., 2023).

For each task, we report the performance of HOMER applied on plain Llama. To further emphasize that our method is orthogonal to the positional encoding scaling methods, and can be applied on top of them, we additionally show the performance of HOMER combined with the best-performing baseline for each task.

### 4.1 PASSKEY RETRIEVAL

In this section, we investigate if HOMER can effectively leverage the long context to handle downstream tasks. We evaluate this on the passkey retrieval task, originally proposed by Mohtashami & Jaggi (2023). In this task, the model is asked to retrieve a random number (called *passkey*) hidden inside distracting texts. The task is widely used to evaluate the maximum context length that the model can effectively handle.

To evaluate the performance at different input lengths, we evaluate the models with inputs of lengths 4k, 8k, 16k, and 32k tokens. We report the retrieval accuracy in Table 1. The result demonstrates that HOMER successfully maintains a high accuracy of around $80\%$ for context length up to 32k tokens which is 8 times longer than the pre-trained context length, while significantly outperforming every baseline. Furthermore, it is also evident that the performance can be further improved by applying HOMER on top of YaRN, the best-performing baseline.

Table 1: Retrieval accuracy on passkey retrieval. Average accuracy on 500 samples are reported. The best values are in **bold**, and the second-best values are underlined. Empty values indicate NaN.

| Method | Context limit | *Llama-2-7b-chat* | | | | *Llama-2-13b-chat* | | | |
|---|---|---|---|---|---|---|---|---|---|
| | | 4K | 8K | 16K | 32K | 4K | 8K | 16K | 32K |
| Plain | 4k | **1.000** | 0.000 | - | - | **1.000** | 0.000 | 0.000 | 0.000 |
| Plain + HOMER | None | 0.990 | **0.924** | **0.890** | **0.776** | **1.000** | **0.944** | **0.882** | **0.804** |
| PI | 8k | 0.432 | 0.356 | 0.000 | - | 0.600 | 0.544 | 0.000 | 0.000 |
| | 16k | 0.006 | 0.006 | 0.006 | 0.000 | 0.022 | 0.028 | 0.018 | 0.000 |
| NTK | 8k | 0.812 | 0.000 | 0.000 | 0.000 | 0.866 | 0.000 | 0.000 | 0.000 |
| | 16k | 0.516 | 0.652 | 0.000 | 0.000 | 0.626 | 0.692 | 0.000 | 0.000 |
| | 32k | 0.106 | 0.194 | 0.162 | 0.000 | 0.286 | 0.570 | 0.442 | 0.000 |
| YaRN | 8k | **0.996** | 0.002 | 0.000 | - | **1.000** | 0.464 | 0.000 | 0.000 |
| | 16k | 0.844 | 0.756 | 0.000 | 0.000 | 0.980 | 0.952 | 0.214 | 0.000 |
| | 32k | 0.702 | 0.654 | 0.696 | 0.002 | 0.926 | 0.888 | 0.836 | 0.026 |
| | 64k | 0.678 | 0.358 | 0.148 | 0.026 | 0.902 | 0.826 | 0.364 | 0.224 |
| YaRN + HOMER | None | **0.996** | **0.984** | **0.876** | **0.802** | **1.000** | **1.000** | **0.974** | **0.860** |

## 4.2 QUESTION ANSWERING

In this section, we push HOMER further and evaluate its performance on a more challenging task: question answering based on long documents. To this end, we measure the model's performance on the validation set of QuALITY (Pang et al., 2021).

For baselines with limited context length, the input documents are clipped to fit in the context limit. For NTK and YaRN, we further clip the documents to be 3/4 of their context limit, as they are only capable of handling inputs slightly shorter than the claimed context limit. This observation can also be found in Section 4.1, as NTK and YaRN models could not handle inputs that are as long as their context limit. For HOMER experiments, we feed the full context into the model as HOMER has no hard limit on the maximum context length.

Table 2: Accuracy in question answering, as evaluated on the QuALITY validation set. The best results are highlighted in **bold**.

| Method | Context limit | Accuracy |
|---|---|---|
| Plain | 4k | 0.327 |
| Plain + HOMER | None | **0.358** |
| PI | 8k | 0.366 |
| NTK | 8k | 0.379 |
| YaRN | 8k | 0.310 |
| NTK + HOMER | None | **0.388** |

We report the prediction accuracy in Table 2. As evident from the table, HOMER effectively extends the context limit, enjoying over 3% of accuracy gain compared to plain Llama. The performance is further improved when applied on top of the best-performing positional encoding scaling method (NTK), achieving 38.8% accuracy. This demonstrates that language models extended with HOMER could potentially perform more sophisticated reasoning based on the extended context.

## 4.3 LANGUAGE MODELING

In this section, we investigate the language modeling fluency of HOMER using the perplexity metric. To this end, we sample 25 long documents from the PG-19 dataset (Rae et al., 2019) and measure the perplexity on documents truncated to specified evaluation lengths.

The core of our methodology is the compression of long context into short embeddings. Aligned with this premise, perplexity is measured iteratively: preceding contexts are condensed with HOMER, and the perplexity of the subsequent segment is deduced based on these compressed contexts. Throughout this procedure, the initial 4k tokens were evaluated using unmodified models, with subsequent tokens assessed in 2k token increments. In every experiment, the last 100 tokens of the input are treated as a suffix.

As illustrated in Table 3, HOMER maintains minimal perplexity values across long documents spanning up to 64k tokens. A more fine-grained perplexity plot is provided in Appendix F. While all other methods either suffer from significant degradation beyond certain thresholds (attributed to

Table 3: Perplexity of 25 long documents from PG-19 truncated to the evaluation length. The best values are in **bold**, and the second-best values are underlined. Empty values indicate NaN.

| Method | Context limit | *Llama-2-7b* | | | | | *Llama-2-13b* | | | | |
|---|---|---|---|---|---|---|---|---|---|---|---|
| | | 4K | 8K | 16K | 32K | 64K | 4K | 8K | 16K | 32K | 64K |
| Plain | 4k | **6.72** | - | - | - | - | 6.14 | $> 10^2$ | $> 10^3$ | $> 10^3$ | $> 10^3$ |
| Plain + HOMER | None | **6.72** | **7.29** | **7.78** | **8.43** | **9.64** | **6.13** | **6.60** | **6.87** | **7.13** | **7.59** |
| PI | 8k | 7.91 | 8.19 | - | - | - | 6.96 | 7.19 | $> 10^2$ | $> 10^3$ | $> 10^3$ |
| | 16k | $> 10$ | $> 10$ | $> 10$ | - | - | $> 10$ | $> 10$ | $> 10$ | $> 10^2$ | $> 10^3$ |
| NTK | 8k | 6.97 | $> 10$ | $> 10^2$ | - | - | 6.26 | 9.62 | $> 10^2$ | $> 10^3$ | $> 10^3$ |
| | 16k | 7.59 | 7.95 | $> 10$ | $> 10^2$ | $> 10^3$ | 6.76 | 7.05 | $> 10$ | $> 10^3$ | $> 10^3$ |
| | 32k | 8.42 | 8.97 | 9.76 | $> 10$ | $> 10^2$ | 7.42 | 7.90 | 8.45 | $> 10$ | $> 10^3$ |
| YaRN | 8k | **6.79** | 7.40 | - | - | - | **6.19** | 6.59 | $> 10^2$ | $> 10^3$ | $> 10^3$ |
| | 16k | 7.00 | 7.32 | 8.98 | - | - | 6.36 | 6.65 | 7.83 | $> 10^2$ | $> 10^3$ |
| | 32k | 7.50 | 8.05 | 8.78 | $> 10$ | - | 6.65 | 7.05 | 7.40 | 8.85 | $> 10^2$ |
| | 64k | 8.49 | $> 10$ | $> 10$ | $> 10$ | $> 10$ | 7.17 | 8.32 | $> 10$ | $> 10$ | $> 10$ |
| YaRN + HOMER | None | **6.79** | **7.09** | **7.52** | **7.95** | **8.83** | **6.19** | **6.51** | **6.78** | **7.02** | **7.44** |

lower scaling factors) or show heightened perplexity even within shorter contexts, HOMER steadily maintains minimal perplexity across extended contexts. This suggests that HOMER is the only method that maintains reasonable fluency even when conditioned on very long contexts. Moreover, HOMER can be seamlessly integrated with conventional positional encoding scaling techniques to further improve performance. As evident from Table 3, applying HOMER on top of YaRN yields lower perplexity.

## 4.4 ABLATION STUDIES

Table 4: Ablation on different components. We report the passkey retrieval accuracy for 500 samples, evaluated on 16k contexts. The best values are highlighted in **bold**.

(a) Token pruning criteria.

| Method | Accuracy |
|---|---|
| Random | 0.006 |
| Attention-based top-K | 0.056 |
| + calibration | **0.890** |

(b) Lower-layer embedding refinement.

| Method | Accuracy |
|---|---|
| No refinement | 0.002 |
| Random | 0.116 |
| Layer-wise top-K | 0.040 |
| Propagative refinement | **0.890** |

In this section, we demonstrate the effectiveness of the design choices made for our method. Specifically, we focus on (1) the proposed token pruning criteria and (2) the method for refining the lower-layer embeddings after applying hierarchical merging. Following the settings of Section 4.1, we compared the retrieval accuracy of each candidate.

**Effectiveness of our token pruning method.** To reduce redundant tokens in the intermediate Transformer layers, we define a calibrated significance score based on the attention weights each token receives from the last token in the chunk. In the pruning procedure, K tokens with the lowest significance scores are dropped. We demonstrate the effectiveness of this criteria by comparing it to a simple baseline, which randomly selects which token to drop. We additionally report the performance with uncalibrated pruning criteria to further emphasize the effectiveness of significance weight calibration. As illustrated in Table 4a, the use of attention-based significance scores and calibration provide an effective proxy for determining the importance of given tokens.

**Effectiveness of propagative refinement.** Another key component of our method is the refinement of the lower-layer embeddings, described as *propagative refinement*. To evaluate its effectiveness, we compare its performance with three alternative approaches (i) not refining the lower-layer embeddings, (ii) gathering random tokens, and (iii) gathering tokens according to their significance score at each layer. As illustrated in Table 4b, propagative refinement achieves the best performance. We credit this to the ability of upper transformer layers to understand high-level information, with their attention weights successfully representing token significance. By selectively providing more significant tokens, propagative pruning reduces computation while improving performance.

Table 5: **Peak memory usage for long inputs.** All measurements are taken on a single A100 GPU, with Flash Attention 2 (Dao, 2023) applied. The baselines include plain Llama, PI, NTK, and YaRN. We report a single value for all baselines as they share the same memory requirement.

| Setup | Peak GPU memory (GB) | | | | |
|---|---|---|---|---|---|
| | 4k | 8k | 16k | 32k | 64k |
| Baselines | 18.2 | 23.6 | 34.4 | 56.7 | > 80 |
| HOMER | 16.3 (-10.8%) | 16.5 (-30.1%) | 16.7 (-51.4%) | 17.6 (-68.9%) | 21.3 (at least -73.4%) |
| + Baselines | 19.2 (+5.5%) | 20.1 (-14.6%) | 20.3 (-41.1%) | 20.8 (-63.4%) | 22.5 (at least -71.8%) |

## 4.5 COMPUTATIONAL EFFICIENCY

In this section, we discuss the computational efficiency offered by our methodology, with a primary focus on memory efficiency. We first demonstrate the computational efficiency of our method by measuring the peak GPU memory usage while processing long inputs. In the following part of the section, we discuss the four key mechanisms that bring efficiency gains.

The peak memory usage for HOMER and baselines is illustrated in Table 5. Note that we report a single number for all baselines (Plain Llama, PI, NTK, YaRN) because the baselines only modify the positional encoding, making no difference in the peak GPU memory usage. For a fair comparison, all methods are tested with Flash Attention 2 (Dao, 2023) enabled. As shown in the table, HOMER significantly reduces memory requirements, reducing the peak memory usage by over 70% when running inference on 64k inputs.

The first source of our efficiency gains is the chunking mechanism. We circumvent the quadratic computation associated with self-attention by processing each chunk separately at the earlier layers. Token reduction is our second source of computation reduction. As our algorithm progressively reduces the number of tokens, fewer tokens have to be processed in the upper layers, reducing the computational overhead. The third source of computation saving is that HOMER outputs concise embeddings, optimizing the subsequent self-attention computation during the generation phase. Compared to naïve forwarding of the complete input, our compact embeddings significantly minimize the size of kv-cache, thus optimizing the computation process. Finally, the memory requirement is further reduced from linear to logarithmic with respect to the input length, thanks to the optimized computation ordering described in Section 3.3. Additional discussion on the inference speed is provided in Appendix E.

## 5 CONCLUSION

In this paper, we introduced Hierarchical cOntext MERging (HOMER), a novel method that efficiently addresses the context limit issue inherent in large language models (LLMs). By employing a strategic divide-and-conquer technique, HOMER prunes redundant tokens, creating compact embeddings while maintaining the richness of information. This approach, validated by our experiments, has proven to be memory-efficient and effective, enabling the handling of extended contexts up to 64k tokens with significantly reduced memory requirements.

**Limitations and future work.** Although our work focuses on training-free extension of context limit, there is no fundamental limit on our method making finetuning impossible. We believe that further improving our method with small-data finetuning can additionally boost performance, and the resulting model would enjoy both the extended context limit and reduced memory requirements.

## ETHICS STATEMENT

While large language models (LLMs) have become a new paradigm of AI academia and business industry by showing remarkable contributions, their critical limitations still remain such as hallucination, biased content generation, and unintended toxicity. The proposed research on long context windows does not address this limitation directly. The influences of longer context limits on the LLM limitations should be more explored, which is one of significant future research directions.

## REPRODUCIBILITY STATEMENT

We outline the implementation details (including detailed algorithm, prompt design, and hyper-parameters) and experiment setups (including tasks, datasets, and metrics) in Section 4 and Appendix B. We also release the source codes.

## ACKNOWLEDGEMENTS AND DISCLOSURE OF FUNDING

This work was supported by Institute of Information & communications Technology Planning & Evaluation (IITP) grant funded by the Korea government (MSIT) (No.2019-0-00075, Artificial Intelligence Graduate School Program (KAIST); No.2021-0-02068, Artificial Intelligence Innovation Hub; No.2022-0-00959, Few-shot Learning of Casual Inference in Vision and Language for Decision Making).

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

APPENDIX

## A MEMORY-EFFICIENT COMPUTATION ORDER

In this section, we outline the proof for the logarithmic memory requirement of the proposed memory-efficient computation ordering suggested in Section 3.3.

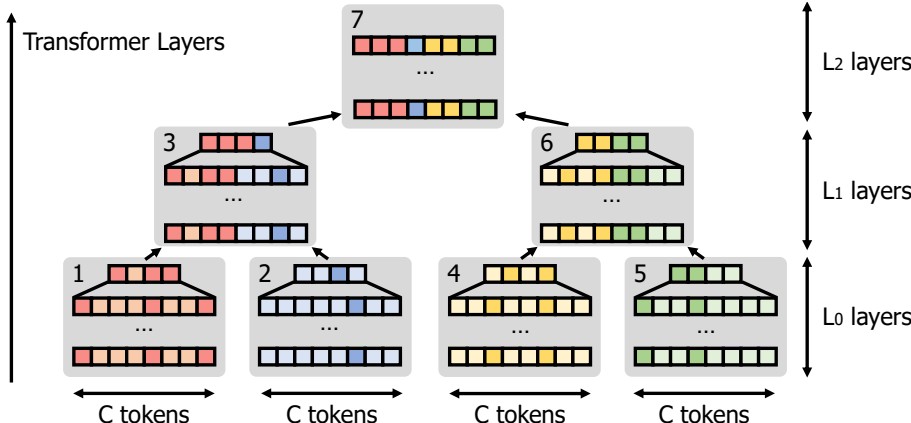

Figure 3: Hierarchical context merging process conceptualized as a binary tree. The top-left numbers of each node denote the memory-efficient computation order. Note that propagative refinement must be applied after processing each node to enjoy the optimized memory usage.

### A.1 PRELIMINARIES

**Problem setup.** We conceptualize the hierarchical context merging process as a binary tree. For example, Figure 3 illustrates a merging process with 4 input chunks.

**Constants.** $L_i$ refers to the number of layers used for processing a chunk at binary tree height $i$. $L := \sum L_i$ is the total number of network layers. $C$ is the maximum chunk size. $M$ is the memory required for storing a key-value pair for a single token in a single layer.

**Remarks.** As the chunk size is bounded, the memory required for forwarding a single chunk through a single layer can be treated as constant. Therefore, it suffices to consider the memory required for storing the key-value pairs at each layer.

Let `FinalMem`($h$) be the memory occupied after processing a binary tree of height $h$. As propagative refinement reduces the intermediate hidden states to be $C/2$ tokens long, `FinalMem`($h$) is bounded as follows.

$$\texttt{FinalMem}(h) = \frac{C}{2} \times \sum_{i=0}^{h} L_i \times M \le \frac{1}{2} LCM$$

### A.2 PROOF

**Proposition.** Let `PeakMem`($h$) be the peak memory usage for processing a binary tree of height $h$. Then,

$$\texttt{PeakMem}(h) \le \left( \frac{1}{2}h + 1 \right) LCM.$$

We prove the proposition using induction. First, consider the leaf node where $h = 0$. As $C$ tokens are passed through $L_0$ layers, the peak memory usage is given as follows, proving the base case.

$$\texttt{PeakMem}(0) = L_0 CM \le LCM$$

Now consider a non-leaf node with $h > 0$. Processing of a non-leaf node consists of three steps: (1) sequentially processing two child nodes, (2) obtaining a merged chunk and forwarding it through $L_h$ layers, (3) applying propagative refinement on the $\sum_{i=1}^{h} L_i$ lower-layer hidden states. As step

(3) is a memory reduction step, `PeakMem`($h$) is the maximum of peak memories of steps (1) and (2).

In step (1) the two child nodes are sequentially processed, resulting in the peak memory of

$$\texttt{FinalMem}(h-1) + \texttt{PeakMem}(h-1).$$

In step (2) hidden states of length $C$ must be held for $\sum_{i=0}^{h} L_i$ layers, so the peak memory is

$$C \times \sum_{i=0}^{h} L_i \times M.$$

By applying the induction hypothesis, we get

$$\texttt{PeakMem}(h) = \max\left\{\texttt{FinalMem}(h-1) + \texttt{PeakMem}(h-1), C \times \sum_{i=0}^{h} L_i \times M\right\}$$

$$\leq \max\left\{\frac{1}{2}LCM + \left(\frac{1}{2}(h-1)+1\right)LCM, LCM\right\}$$

$$= \max\left\{\left(\frac{1}{2}h+1\right)LCM, LCM\right\}$$

$$= \left(\frac{1}{2}h+1\right)LCM.$$

As the peak memory grows linearly with the tree height, it grows logarithmic with the input sequence length.

## B  IMPLEMENTATION DETAILS

**Context merging schedule.** Following the formulation in Figure 3, we detail how many layers are assigned to each level of the binary tree. The basic principle is to assign an equal number of layers to each node. In practice, we noticed that additionally assigning more layers to the leaf nodes helps improve the overall performance. We assign 12 additional layers for 7b models and 20 layers for 13b models.

**Calibration.** For all models (HOMER and HOMER+baselines), calibration is performed using 100 text corpora segments from the validation set and the test set of WikiText-103 (Merity et al., 2016).

**Maximum chunk length.** In all experiments involving HOMER, the maximum chunk length was set to be half of the context limit.

## C PROMPTS FOR DOWNSTREAM TASKS

The detailed prompt format for each downstream task are provided in Table 6 and Table 7.

Table 6: **Prompt for passkey retrieval task.** Slight modifications are made from the original prompt to turn it into a chat prompt (Touvron et al., 2023).

| | |
|---|---|
| Prefix | **[INST]** <<**SYS**>>
There is an important info hidden inside a lot of irrelevant text.
Find it and memorize them. I will quiz you about the important information there.
<<**/SYS**>> |
| Context | The grass is green. The sky is blue. The sun is yellow. Here we go. There and back again. (repeat x times)
The pass key is 12323. Remember it. 12323 is the pass key.
The grass is green. The sky is blue. The sun is yellow. Here we go. There and back again. (repeat y times) |
| Suffix | What is the pass key? The pass key is
**[/INST]** |

Table 7: **Prompt for question answering task.** Basic prompt format follows Shaham et al. (2023). Slight modifications are made to turn it into a chat prompt (Touvron et al., 2023).

| | |
|---|---|
| Prefix | **[INST]** <<**SYS**>>
You are provided a story and a multiple-choice question with 4 possible answers (marked by A, B, C, D). Choose the best answer by writing its corresponding letter (either A, B, C, or D). Do not provide any explanation.
<<**/SYS**>> |
| Context | (The actual document) |
| Suffix | Question and Possible Answers:
{question}
(a) {choice 1}
(b) {choice 2}
(c) {choice 3}
(d) {choice 4}
Answer:
**[/INST]** |

## D    ILLUSTRATION OF PROPAGATIVE REFINEMENT

In this section, we provide a comprehensive explanation of propagative refinement suggested in Section 3.2. Figure 4 illustrates the process, where 3 out of 6 tokens are pruned at layer N.

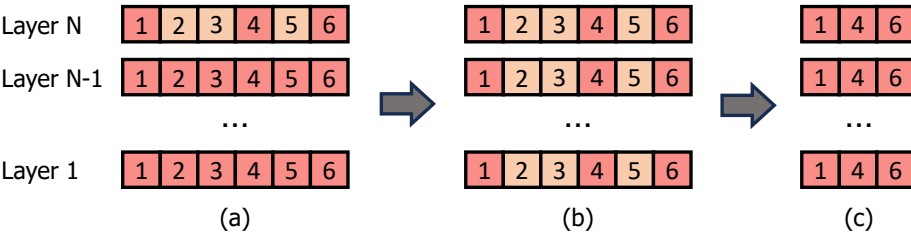

Figure 4: Illustration of the propagative refinement process.

Initially, as shown in part (a), the three least significant tokens (2, 3, and 5) are marked for pruning in layer N. Subsequently, in part (b), the corresponding tokens in the lower-layer embeddings are also marked for pruning. Finally, part (c) demonstrates the outcome after pruning, where all marked tokens are eliminated across every layer, resulting in a uniform, compressed embedding structure composed of just three tokens.

## E    INFERENCE SPEED ANALYSIS

In this section, we discuss the inference speed of HOMER. Besides reducing memory requirements, HOMER also provides a significant speedup due to the extensive reduction in computation. Table 8 illustrates the average inference time for HOMER and other baselines. Specifically, we compare the time required to generate 20, 50, and 100 tokens, conditioned on 8k, 16k, and 32k contexts.

Table 8: **Inference time for long inputs.** All measurements are taken on a single A100 GPU, with Flash Attention 2 (Dao, 2023) applied. We also report the percentage of the speedup. We report a single value for all baselines[*] following the setup in Section 4.5. The baselines include plain Llama, PI, NTK, and YaRN. The inference time is averaged over 25 runs.

| Setup | Average run time (seconds) | | |
| --- | --- | --- | --- |
| | 8k | 16k | 32k |
| *20 tokens* | | | |
| Baselines* | 1.879 | 3.224 | 6.546 |
| HOMER | 1.673 (12.3% speedup) | 2.270 (42.0% speedup) | 3.513 (86.3% speedup) |
| *50 tokens* | | | |
| Baselines* | 3.842 | 6.028 | 11.143 |
| HOMER | 3.026 (27.0% speedup) | 3.639 (65.6% speedup) | 4.873 (128.7% speedup) |
| *100 tokens* | | | |
| Baselines* | 7.149 | 10.733 | 18.828 |
| HOMER | 5.355 (33.5% speedup) | 5.930 (81.0% speedup) | 7.169 (162.6% speedup) |

The main source of performance gain in HOMER, as described in Section 4.5, is the computation reduction. The following points highlight these improvements:

- The divide-and-conquer approach circumvents the quadratic computation associated with self-attention.
- Token pruning significantly reduces the number of tokens to process, especially in the upper layers.
- HOMER compresses long contexts into short embeddings, substantially reducing the size of the kv-cache. This step lowers the computational demand during the decoding stage.

As evident from the results, HOMER provides a significant speedup (up to 162.6%) compared to the baseline methods. It's important to note that our method is even more beneficial when generating longer outputs conditioned on longer inputs, underscoring its effectiveness in handling long contexts.

We also emphasize that the additional computation introduced by HOMER is minimal. The hierarchical context merging process involves relatively cheap operations, including matrix subtraction (calibration), gathering by index (calibration, token pruning, and propagative refinement), top-k selection (token pruning), and tensor concatenation (merging). Conversely, it reduces more costly operations such as matrix multiplication for computing self-attention.

## F  PERPLEXITY PLOT FOR LANGUAGE MODELING EXPERIMENT

In this section, we provide a fine-grained perplexity plot for the fluency experiment in Section 4.3.

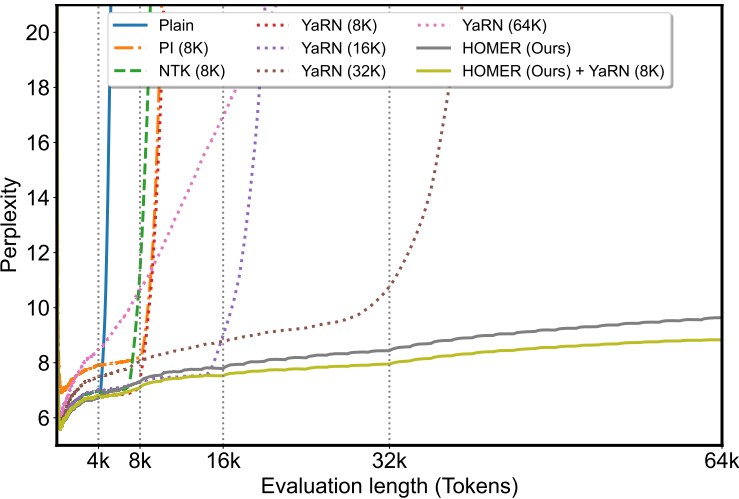

Figure 5: Perplexity plot on 25 long documents from PG-19 dataset (Rae et al., 2019), measured with Llama-2-7b. HOMER consistently achieves low perplexity across long documents up to 64K tokens, demonstrating its ability to remain fluent while conditioned on very long inputs. Detailed comparison with more baselines are provided in Table 3.

# G PERPLEXITY EVALUATION ON DOWNSTREAM TASKS

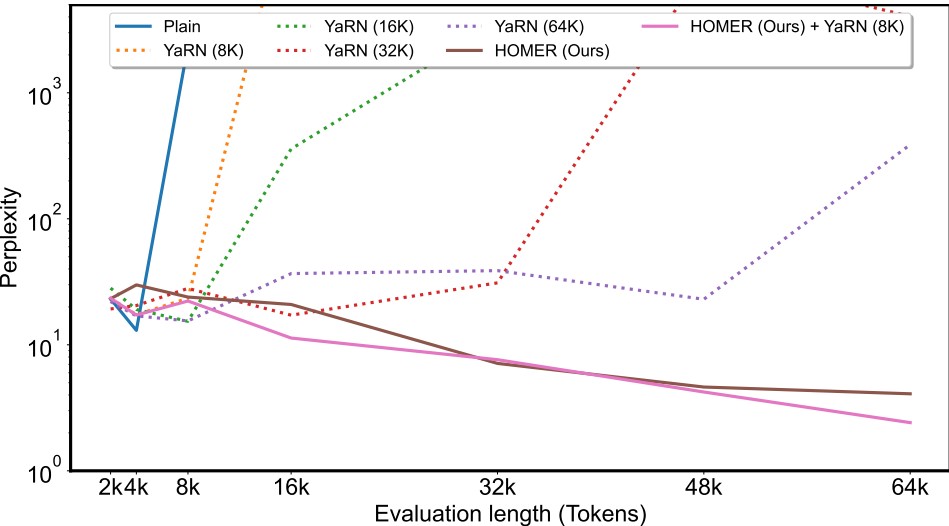

Figure 6: Perplexity plot on 100 long samples from passkey retrieval, measured with Llama-2-7b-chat. HOMER achieves lower perplexity when conditioned on longer inputs, demonstrating its ability to effectively handle the long inputs. Perplexity values on landmark lengths are provided in Table 9.

Table 9: **Perplexity values for passkey retrieval.** Empty values indicate NaN.

|  | Context Limit | 2k | 4k | 8k | 16k | 32k | 64k |
|---|---|---|---|---|---|---|---|
| Plain | 4k | 23.12 | 13.013 | $> 10^3$ | - | - | - |
| Plain + HOMER | None | 23.22 | 29.882 | 23.932 | 20.897 | 7.119 | 4.085 |
| YaRN | 8k | 23.414 | 17.338 | 23.394 | $> 10^4$ | - | - |
|  | 16k | 28.183 | 19.07 | 15.366 | $> 10^2$ | $> 10^3$ | - |
|  | 32k | 19.248 | 20.491 | 28.034 | 17.187 | 31.017 | $> 10^3$ |
|  | 64k | 22.031 | 16.963 | 15.481 | 36.760 | 38.786 | $> 10^2$ |
| YaRN + HOMER | None | 23.414 | 17.232 | 22.263 | 11.317 | 7.618 | 2.412 |

In this section, we provide additional perplexity experiments on a more challenging benchmark where accessing previous long contexts is essential. To achieve this, we reformulated the passkey retrieval task in Section 4.1 and measured the perplexity of ground-truth answer phrases (e.g., 'The passkey is 12321.'). The results are demonstrated in Figure 6 and Table 9.

As the results show, HOMER exhibits lower perplexity when conditioned on longer contexts, achieving its best performance with 64k inputs. Furthermore, HOMER outperforms the long-context competitors with context lengths of 16k and beyond. These experiments emphasize the efficacy of HOMER in utilizing long contexts, particularly in scenarios where accessing such context is necessary.

