# OpenReview forum: "Hierarchical Context Merging: Better Long Context Understanding for Pre-trained LLMs"
_ICLR.cc/2024/Conference — ICLR 2024 poster_

### Official Review · Reviewer_wcXN · 2023-10-27

**Soundness:** 4 excellent
**Presentation:** 3 good
**Contribution:** 4 excellent
**Rating:** 8
**Confidence:** 4

**Summary:**

Paper addresses long contexts (>4k tokens of a pre-trained model such as LLaMA) in LLMs which are challenging because of the nature of Transformer self-attention.
A technique called HOMER is introduced: contrarily to previous approaches that approximate the computation of self attention, or that use position interpolation at inference, authors propose a ‘compression’ method which decomposes the long input into manageable chunks; then a hierarchical merging strategy ensures the ‘communication’  between those chunks. One advantage of the approach is that it does not necessarily require fine-tuning. Perplexity experiments are done to assess the efficiency of the method as well as experiments on passkey retrieval and Q&A tasks. Finally it is shown that the approach proposed can be used successfully in addition to position interpolation techniques.

**Strengths:**

-a new approach to handling long contexts in LLMs that can be used additionally to position interpolation techniques proposed recently

-extensive experiments done with LLaMA2 (ppl and long-context tasks) and comparison with recent (2023) baselines such as YaRN

-convincing results on both ppl and tasks evaluations

-method is orthogonal to position interpolation methods

**Weaknesses:**

-the parts related to propagative refinement (3.2) and efficiency improvement (3.3) should be clarified/improved

-for Q&A it is a pity that HOMER+YaRN results are not presented (+ YaRN seems to not be working well on this task, why?)

**Questions:**

-It is unclear to me why PROPAGATIVE REFINEMENT OF LOWER-LAYER EMBEDDINGS is needed. During inference the final layer representation of the long input is computed so why do you actually need to refine the lower layer embeddings afterwards ? In other words, why do you need to have fixed-length embeddings for every layers afterwards ? I could not fully understand this part.

-Similarly section 3.3 is a bit to laconic to be fully understandable: how this binary-tree inspired computation is innovative ? What is described in Appendix/figure3 seems to me a straightforward way to efficiently compute the hierarchical merging process.

-for Q&A it is a pity that HOMER+YaRN results are not presented (+ YaRN seems to not be working well on this task, why?)

-for Q&A i’m wondering if the comparison with YaRN and NTK is really fair as for them, context documents are clipped whereas for HOMER, which is a compression technique, the full context document can be used.

-In 4.4 (ablation) it not 100% clear what the uncalibrated (as opposed to calibrated) pruning criteria is

---

> ### Author Response · Authors · 2023-11-16
> **Response to Reviewer wcXN (1/2)**
>
> Dear reviewer wcXN,
>
> We sincerely appreciate your efforts in reviewing our manuscript. We respond to each comment in the following content. We carefully incorporated the discussions into the revised manuscript. We highlighted the revised contents in blue for your convenience to check.
>
> ---
> **[W1] The parts related to propagative refinement (3.2) and efficiency improvement (3.3) should be clarified/improved**
>
> We have revised the writing in Section 3.2, and added a step-by-step example in Appendix D to aid the readers’ understanding. We have also revised Section 3.3, elaborating why our computation order is innovative.
>
> In section 3.2, we provided an alternative explanation of the propagative refinement process. We described the process as an additional refinement step taken after token reduction. When a token is pruned in the upper layers, the corresponding tokens (i.e. tokens that originated from the same input token) are also pruned in the lower-layer embeddings. This ‘propagates’ the pruning decision of the upper layers to the lower layers embeddings, hence the name. The synchronized pruning across layers results in shorter, uniform embeddings for each layer.
>
> For example, consider the situation where the tokens at positions 2, 3, and 5 are pruned in the N-th layer. By applying propagative refinement, we simply drop the tokens at positions 2, 3, and 5 from the lower-layer embedding, too.
>
> For changes in section 3.3, please refer to Q2.
>
> ---
> **[W2] HOMER+YaRN results are not presented for Q&A (+ YaRN seems to not be working well on this task, why?)**
>
> We conducted additional experiments of HOMER+YaRN for Q&A. HOMER+YaRN achieves an accuracy of 0.360, improving the performance of YaRN (0.310). Since we focused on demonstrating HOMER's applicability in enhancing the best-performing baselines, only HOMER+NTK was explored in the paper.
>
> YaRN introduces multiple hyperparameters, including the temperature t required for scaling the attention weights, and alpha/beta values required for applying NTK-by-parts. We believe that the suggested choice of the hyperparameters can be suboptimal for some tasks, as they were empirically searched by minimizing the perplexity.
>
> ---
> **[Q1] Why do we need to reduce the lower layer embeddings, and why do the embeddings have to be the same length for every layer?**
>
> This is because our method focuses on compressing the long context into shorter embeddings, reducing the computational burden of handling long context. Without refinement, the lower layer embeddings would remain large, resulting in large memory requirements and computational overhead in future decoding steps. Therefore, we further reduce the lower layer embeddings with propagative refinement.
>
> We decided to make the embedding dimension the same in every layer, in order to make it more compatible with standard KV-cache (which typically has fixed length across all layers).
>
> ---
> **[Q2] Section 3.3 requires more explanation. How is binary-tree-inspired computation innovative? It seems like a straightforward way to efficiently compute the hierarchical merging process.**
>
> In typical Transformer models, all tokens at a single layer are computed in parallel. A similar implementation of HOMER would involve processing multiple chunks concurrently. The memory consumption can be large in this implementation, although it will already require less GPU memory than the baselines thanks to token reduction of HOMER.
>
> Section 3.3 introduces an alternative computation ordering for HOMER, where chunks are processed sequentially rather than in parallel. This sequential handling of chunks, made feasible by our divide-and-conquer approach, significantly lowers GPU memory usage. We have refined the manuscript to more clearly convey this aspect.
>
> It is also important to note that while the sequential computation may slightly increase the inference time, HOMER provides speedup (up to 162.6%) even with the memory-efficient computation order, thanks to its substantial computation reduction.
>
> ---
> **[Q3] For Q&A it is a pity that HOMER+YaRN results are not presented (+ YaRN seems to not be working well on this task, why?)**
>
> Please refer to W2.
>
> ---
> **[Q4] Is comparison with YaRN and NTK fair for the QA tasks? Context documents are clipped for them, whereas HOMER uses the full context.**
>
> When YaRN and NTK are further scaled to have 16k context limit (which enables them to get full context documents as inputs), both of their performance degrade (YaRN scores 0.233 and NTK scores 0.369). Therefore we reported the highest score achievable by YaRN and NTK, making the comparison fair. This aligns with the observation in the language modeling experiments (Section 4.1), where the performance of RoPE scaling methods significantly degrades with larger scaling factors. We believe that in the QA tasks, such performance degradation outweighs the benefit of seeing longer context.

---

> > ### Comment · Reviewer_wcXN · 2023-11-19
> > **response after authors' feedback**
> >
> > dear authors
> >
> > tks for your answers and for adding experiments on HOMER+YaRN

---

> > > ### Author Response · Authors · 2023-11-20
> > > **Thank you for your response**
> > >
> > > Dear Reviewer wcXN,
> > >
> > > Thank you for your comments and for taking the time to review our manuscript.
> > >
> > > We are happy that our additional discussion and experiments addressed your concerns!
> > >
> > > If you have any further comments or suggestions, please let us know. We are committed to improving the quality of our work, and we value your feedback.
> > >
> > > Thank you very much,
> > >
> > > Authors

---

> ### Author Response · Authors · 2023-11-16
> **Response to Reviewer wcXN (2/2)**
>
> **[Q5] What does ‘uncalibrated’ mean in section 4.4?**
>
> It means that the calibration technique described in section 3.1 (second paragraph of ‘Token reduction on individual chunks’) is not applied. In other words, the raw attention logits are used as significance scores for pruning.

---

### Official Review · Reviewer_ae6y · 2023-10-28

**Soundness:** 2 fair
**Presentation:** 3 good
**Contribution:** 2 fair
**Rating:** 3
**Confidence:** 3

**Summary:**

This paper proposes an approach to extend the context limit of large language models (LLMs). The key idea is to hierarchical context merging, progressively conducting token pruning from bottom to top layers. The token pruning method is simple and training-free, which prunes the tokens with minimal attention from the final token in each chunk, and conducts adjustments (e.g., attention and position bias) to make sure the model to work properly. The results show that it can work on 64k context length without perplexity explosion.

**Strengths:**

- A training-free method and can be easily adopted
- Some empirical results showing the feasibility of the token pruning method.

**Weaknesses:**

While this paper presents a seemingly promising approach to long context modeling, its major issue lies in the lack of clarity regarding its use case. According to the paper's description, HOMER first breaks down a long context (e.g., 64k) into N chunks, prunes tokens from each chunk, and finally merges them into a compressed and short context. If this is the case, this method is not suitable for streaming decoding scenarios (where the key-value cache of previously decoded tokens is not recalculated, only new decode tokens are computed).

The key question arises: if this work is not for streaming decoding, then there is no need for such a process. For a 64k length context, wouldn't it be sufficient to ignore the first 62k context and only use the last 2k context? According to the results in Figure 1 of the paper, using only 2k context, the perplexity (PPL) is less than 6 (around 5 point something), whereas after all the effort of merging with HOMER, the PPL is above 8.

Given this, I question the necessity of using HOMER. Using the last 2K context directly seems to yield better results, casting a large doubt on the value of the method proposed in this paper. If the setting of this paper is not for streaming decoding, then the performance of HOMER should at least be better than the PPL of the 2K context. However, this method requires recalculating the key-value pairs for the previous context, which doesn't seem to align with a streaming decoding setting, making it the most confusing aspect for me.

**Questions:**

See the weakness section

---

> ### Author Response · Authors · 2023-11-16
> **Response to Reviewer ae6y**
>
> Dear reviewer ae6y,
>
> We sincerely appreciate your efforts in reviewing our manuscript. We respond to each comment in the following content. We carefully incorporated the discussions into the revised manuscript. We highlighted the revised contents in blue for your convenience to check.
>
> ---
> **[W1] It seems that HOMER cannot be used in the streaming decoding scenario.**
>
> We clarify that HOMER is compatible with streaming decoding, as described in Section 3.2 (‘Using the refined embeddings for further generation’), and all of our experiments involving generation (Sections 4.2 and 4.3) are performed with streaming decoding. HOMER compresses the long input (i.e. the ‘prompt’) and produces fixed-length embeddings, which are then utilized for subsequent decoding steps, just like a key-value cache. The decoding process is streamlined; the HOMER embeddings and the key-value cache of previously decoded tokens do not undergo recalculation at every decoding step - the hierarchical context merging happens only once at the beginning.
>
> For a more comprehensive explanation, let us first illustrate the process of a typical streaming decoding scenario.
> 1. Forward the full prompt to generate key-value cache
> 2. Iteratively decode new tokens, adding the new keys and values into the cache.
>
> In case of HOMER, step 1 is simply replaced with hierarchical context merging to create compact embeddings (or effectively the ‘key-value cache’) that can be used for further decoding process.
>
> ---
> **[W2] Using the last 2K context directly seems to yield better results.**
>
> We emphasize that by only using the last 2K tokens, long context cannot be properly handled. For example, the model would not be able to answer a question about a book if it can only access the last 2K content. HOMER's approach, despite a higher PPL, is specifically designed to leverage the rich information in longer contexts, which is not possible with only the last 2K tokens. Our experiments in pass key retrieval and question answering demonstrate HOMER's proficiency in utilizing the entire context, thereby significantly enhancing the model's performance in tasks requiring comprehensive context understanding.
>
> Additionally, we remind that the perplexity experiments demonstrate that HOMER handles long inputs more effectively compared to other baselines. Therefore, a more appropriate comparison would be between HOMER and other methods at similar context lengths. Our results indicate that HOMER outperforms these models in handling long inputs, which underscores its value in scenarios where the full context is essential.
>
> ---
> **[W3] Given this, I question the necessity of using HOMER.**
>
> In summary, it is a misunderstanding that HOMER cannot be used in streaming decoding, and HOMER is actually intended to be used in such scenarios. Also, HOMER can effectively utilize the information in long contexts as demonstrated in Sections 4.2 and 4.3, which would have been impossible if only the last 2K tokens were used. Given this, we believe that HOMER is indeed a promising approach for long context handling, applicable to practical use cases, efficiently managing and utilizing the long context. We hope our response clarifies your concerns.

---

> > ### Author Response · Authors · 2023-11-21
> > **A Gentle Reminder**
> >
> > Dear Reviewer ae6y,
> >
> > Thank you very much again for your time and efforts in reviewing our paper.
> >
> > We kindly remind that we have only two days or so in the discussion period.
> > We just wonder whether there is any further concern and hope to have a chance to respond before the discussion phase ends.
> >
> > Many thanks,
> >
> > Authors

---

> ### Comment · Reviewer_ae6y · 2023-11-22
>
> Thank you for the response provided by the authors. However, I must express my disagreement with the justification that longer context leading to higher Perplexity (PPL) is acceptable. As the authors mentioned, a 2K context may not handle scenarios that require longer text, which motivates this work that can support/access longer context. However, there is a distinction between merely supporting/accessing a longer context and being able to effectively leverage a longer context. **If longer contexts do not result in lower perplexity, it suggests that the model is not effectively leveraging the extended context, and may even be less effective than using just the last 2k tokens of context.**
>
> Prior to 2023, nearly all research papers focusing on long contexts have justified their contributions by demonstrating lower perplexity with the use of longer contexts. I am puzzled as to why, in 2023, it has become acceptable for work on long contexts to be considered a valid contribution as long as the PPL does not skyrocket to hundreds or thousands. Perhaps I am not keeping pace with the times, but I firmly believe that **if using a longer context does not improve perplexity over using just the latest 2k context window, then what is the significance or practical value of this work?** Unless this work can support streaming decoding, it would be entirely feasible to recalculate the last 2k tokens of context to serve as the context, which could actually result in lower perplexity.
>
> Regarding streaming decoding, I feel that the authors may not have fully understood what I mean by streaming decoding. If, as the authors state, context merging is only performed at the beginning, then the tokens I decode subsequently will not be merged. If I continue to decode thousands or even tens of thousands of tokens thereafter, HOMER will not assist me in merging the context again, which is why I say HOMER doesn't support the streaming decoding setting. If you need to recaculate for the new decoded tokens for online context merging, then it is not the streaming decoding scenario --  Given that the last 2k contexts can yield better PPL, one can recaculate the last 2k tokens as the context so that the PPL can be better.
>
> The authors mention other baselines may also have higher perplexity. However, I hope that the authors check whether these previous papers also don't support the streaming decoding setting where they can support long context without recalculating for the new decoded tokens. For example, the work "EFFICIENT STREAMING LANGUAGE MODELS WITH ATTENTION SINKS" [1] is a typical streaming decoding method -- even if its PPL increases, it still has practical value. (I'm not asking the authors to compare with [1] as they are concurrent work. I mention [1] just as an example to demonstrate my viewpoint regarding justification of higher PPL in the streaming decoding setting)
>
> [1] Guangxuan Xiao, Yuandong Tian, Beidi Chen, Song Han, Mike Lewis: Efficient Streaming Language Models with Attention Sinks. https://arxiv.org/abs/2309.17453

---

> ### Author Response · Authors · 2023-11-23
> **Additional Response to Reviewer ae6y (1/2)**
>
> Dear reviewer ae6y,
>
> Thank you for your comments and for providing us with the opportunity to further address your concerns.
>
> ---
>
> **Further discussions on perplexity**
>
> We now understand your concern better. The increase of perplexity is due to the characteristics of the current benchmark used in our language modeling experiment (PG-19), where one can predict future words without fully understanding previous long contexts. In the following response, we clarify this point in more detail and demonstrate new results on a new benchmark where leveraging long context is more important.
>
> PG-19 serves as a relatively simple benchmark, where the models can predict the next words based on local context. Consequently, both HOMER (ours) or YaRN (baseline) may increase perplexity due to distribution shifts, i.e., handling unseen long contexts, which outweigh their capabilities of understanding long contexts. Nevertheless, we think the experiment remains valuable as they demonstrate that HOMER significantly outperforms the baselines for the same purpose, even accommodating contexts as long as 64k, in terms of fluency (i.e., low perplexity). We have clarified these points in our revised manuscript.
>
> To emphasize the efficacy of understanding long contexts, we conducted additional perplexity experiments on a more challenging benchmark where accessing previous long contexts is essential. To achieve this, we reformulated the passkey retrieval task in Section 4.1 and measured the perplexity of ground-truth answer phrases (e.g., 'The passkey is 12321.'). We included the perplexity plots in Appendix G and present the results in a table here (the empty values indicate NaN.).
>
>
>
> |               | Context Limit | 2k      | 4k      | 8k       | 16k       | 32k       | 48k        | 64k       |
> |---------------|---------------|---------|---------|----------|-----------|-----------|------------|-----------|
> | Plain         | 4k            | 23.12   | 13.013  | $>10^3$ | -         | -         | -          | -         |
> | Plain + HOMER | None          | 23.22   | 29.882  | 23.932   | 20.897    | 7.119     | 4.622      | 4.085     |
> | YaRN          | 8k            | 23.414  | 17.338  | 23.394   | $>10^4$ | -         | -          | -         |
> |               | 16k           | 28.183  | 19.070   | 15.366   | $>10^2$   | $>10^3$  | $>10^3$   | -         |
> |               | 32k           | 19.248  | 20.491  | 28.034   | 17.187    | 31.017    | $>10^4$  | $>10^3$ |
> |               | 64k           | 22.031  | 16.963  | 15.481   | 36.76     | 38.786    | 22.979     | $>10^2$   |
> | YaRN + HOMER  | None          | 23.414  | 17.232  | 22.263   | 11.317    | 7.618     | 4.221      | 2.412     |
>
>
> As the results show, **HOMER exhibits lower perplexity when conditioned on longer contexts**, achieving its best performance with 64k inputs. Furthermore, HOMER outperforms the long-context competitors with context lengths of 16k and beyond. These experiments emphasize the efficacy of HOMER in utilizing long contexts, particularly in scenarios where accessing such context is necessary.
>
> Nevertheless, we agree with your point that current perplexity plots may confuse readers. To further clarify the benefits of HOMER, we revised the organization of our manuscript to emphasize downstream tasks and then discuss perplexity fluency. Additionally, we changed Figure 1 to highlight retrieval performance instead of perplexity. We believe your feedback has sharpened the merits of our work. Thank you!

---

> > ### Author Response · Authors · 2023-11-23
> > **Additional Response to Reviewer ae6y (2/2)**
> >
> > **Generating extremely long output with HOMER**
> >
> > HOMER can generate extremely long outputs by iteratively generating and merging the outputs. For example, HOMER may generate 2k tokens conditioned on 8k context (which is compressed into 2k embeddings) in one iteration. In the next iteration, HOMER can compress the 10k context into 2k embeddings and further generate additional 2k tokens. By repeating this procedure, HOMER can generate extremely long outputs while being able to utilize full information in the inputs and the previously decoded outputs.
> >
> > While it is true that the whole process of long output generation with HOMER is not precisely streaming decoding (as the compressed embeddings have to be recomputed every iteration), we emphasize that the additional cost introduced by recomputation is relatively small. To demonstrate this, we compare the time required for generating compressed embeddings (i.e. HOMER), and time required for the actual generation.
> >
> > In the following table, t_decoding denotes the time required for generating 2K tokens conditioned on pre-computed 2K key-value cache. t_compression denotes the time required for compressing the input of given length (4K, 8K, 16K, 32K, and 64K) into 2K embeddings. All measurements are done on a single A100 GPU.
> >
> >
> > |                | 4K    | 8K    | 16K   | 32K   | 64K   |
> > |----------------|-------|-------|-------|-------|-------|
> > | t_decoding     | 96.08 | 96.08 | 96.08 | 96.08 | 96.08 |
> > | t_compression  | 0.374| 0.665| 1.280 | 2.445 | 5.352 |
> > | % of decoding time | 0.39% | 0.69% | 1.33% | 2.54% | 5.57% |
> >
> >
> > As evident from the above results, **HOMER can be simply extended to generate long outputs, while being able to utilize the full previous context, without introducing a significant computational overhead.**
> >
> > Regarding the comparison with StreamingLLM, we emphasize that StreamingLLM cannot access the previous context due to its limited attention window. Thus, it does not perform well on tasks that demand an understanding of long contexts, such as those involving information retrieval from the contexts [1].
> >
> > [1] https://openreview.net/forum?id=NG7sS51zVF&noteId=UZKrAqkWdI

---

### Official Review · Reviewer_X3So · 2023-10-30

**Soundness:** 2 fair
**Presentation:** 3 good
**Contribution:** 3 good
**Rating:** 6
**Confidence:** 4

**Summary:**

The paper introduces a new method called Hierarchical Context Merging to extend the context limit of LLMs all the way to 64K tokens. The method is training-free, and uses a divide-and-conquer methodology to segment inputs into smaller units. At each following Transformer layer, the segments are fused together through a token reduction technique to reduce the memory footprint. The authors also propose to optimize the computation order to further reduce the memory requirement by 70%.

The proposed method is evaluated on three sets of tasks: Perplexity on 19 long documents from PG-19, passkey retrieval and question answering. The paper shows superior performance on all three benchmarks against RoPE-based baselines such as PI, NTK and YaRN.

**Strengths:**

The work shows several strengths:
1. The proposed method HOMER is the only method which has a reasonable Perplexity on 64K long documents. The performance gain on passkey retrieval at 32K tokens is impressive: 22.4% -> 80.4%.
2.  The memory reduction up to 70% is also quite impressive.

**Weaknesses:**

However, I do have several weaknesses regarding the work:
1. For Llama-2-13B, the gain on both perplexity and passkey retrieval is very marginal for all the context lengths below 32K.
2. The Accuracy on QA benchmark is very concerning (Table 3): HOMER alone is worse than the NTK method, not showing any performance advantage. 1&2 together raises the question of the generalization of the HOMER method, and the presented results are not comprehensive enough to convince the readers.
3. The biggest concern is the complexity of the method: compared to RoPE based methods, HOMER involves merging chunks, token reduction, embedding refinement as well as computational order reduction. All these techniques together make the proposed method very hard for wide adoption in the community, hence limiting the impact of the work.

**Questions:**

I do have a few questions that I need clarifications from the authors:

1. The inference time of HOMER was not discussed at all in the paper. Instead, memory is the major focus. While memory footprint is an important dimension, inference time is very important for LLM applications. Given the extensive manipulations of HOMER on context embeddings at each Transformer layer, I expect a significant overhead occurring due to such forward operations compared to RoPE methods.
2. The explanation of propagative refinement is confusing: token pruning is done through a forward pass, and the top-K token embeddings are tracked and obtained only at the top-layer after the entire forward pass. It is unclear to me how the refined embeddings are gathered at the low-level layers, and impact the computation results shown in the ablation in Table 4(b).
3. Minor point: the token pruning section didn’t explain how it is done – it is unclear how the top-K hyperparameter is selected, and the tradeoff between speed and K is made (e.g. keeping a large K at each layer will result in more steps of pruning).

---

> ### Author Response · Authors · 2023-11-16
> **Response to Reviewer X3So (1/3)**
>
> Dear reviewer X3So,
>
> We sincerely appreciate your efforts in reviewing our manuscript. We respond to each comment in the following content. We carefully incorporated the discussions into the revised manuscript. We highlighted the revised contents in blue for your convenience to check.
>
> ---
> **[W1] For Llama-2-13B, the gain on both perplexity and passkey retrieval is very marginal for context lengths below 32k.**
>
> Our primary goal with HOMER is to extend the context limit of large language models, focusing on very long contexts that were previously unmanageable. Hence, the performance comparisons under long context lengths (e.g., >= 32k) are more important, and the gain of HOMER is **substantial** for both perplexity and passkey retrieval in this regime.
>
> Furthermore, the gain of HOMER is often significant even in the case of shorter context lengths (e.g., < 32k). For example, at 16k pass key retrieval task, we achieve 97.4% accuracy, while the best-performing baseline, YaRN, does 83.6%. Furthermore, we emphasize that HOMER achieves superior performance with greater efficiency (51.4% memory savings and 42.0% speedup at 16k context), which is its important additional advantage.
>
> ---
> **[W2] HOMER alone performs worse than the baselines for multiple-choice QA task.**
>
> It is important to note that compatibility with other RoPE scaling methods is one of our main advantages. Combined with these methods, we **always** outperform the baselines for every experiment on extended context.
>
> We also emphasize that applying HOMER consistently gives performance gain, both when applied on the plain model and NTK. This demonstrates HOMER’s ability to successfully leverage the extended context.
>
> Furthermore, we highlight that even HOMER alone achieved superior performance in most of our experiments. Namely, HOMER alone already showed the best performance among all baselines in 15 out of 19 experiments reported in our paper.
>
> ---
> **[W3] The biggest concern is the complexity of the method, compared to the RoPE scaling methods. This could make HOMER very hard for wide adoption in the community.**
>
> HOMER is very easy-to-use, requiring only one line of code. This is made possible by providing a ‘patch’ function (analogous to [1]), which automatically edits an existing model to use HOMER. We believe that HOMER will be easily adopted by the community, thanks to the simple and straightforward implementation.
>
> We also emphasize that our method is training-free, which is a great advantage in terms of its impact. For example, our current implementation of Llama can already be directly applied to all Llama derivatives (e.g. Vicuna, Alpaca, Koala, etc.) in a plug-and-play manner. This is a significant advantage compared to other long-context methods [2][3][4][5][6][7][8], which necessitate fine-tuning.
>
> Finally, we highlight that our computational complexity is significantly lower than the baselines, providing up to 73.4% memory saving while being 162.6% faster. We believe that the significant efficiency gain will also encourage the wide adoption of HOMER.
>
> ---
>
> [1] https://github.com/jquesnelle/yarn/blob/master/scaled_rope/patch.py
>
> [2] Landmark Attention: Random-Access Infinite Context Length for Transformers, NeurIPS 2023
>
> [3] Augmenting Language Models with Long-Term Memory, NeurIPS 2023
>
> [4] Recurrent Memory Transformer, NeurIPS 2022
>
> [5] Memorizing Transformers, ICLR 2022
>
> [6] In-context Autoencoder for Context Compression in a Large Language Model, under review for ICLR 2024 (scores: 8/8/5/5)
>
> [7] CLEX: Continuous Length Extrapolation for Large Language Models, under review for ICLR 2024 (scores: 8/6/6/5)
>
> [8] PoSE: Efficient Context Window Extension of LLMs via Positional Skip-wise Training, under review for ICLR 2024 (scores: 6/6/6/5)

---

> > ### Comment · Reviewer_X3So · 2023-11-22
> >
> > Thank you authors for addressing my concerns. While I appreciate that the authors provide a one-line plug-and-play code, I still think the implementation complexity of HOMER method is high. However, given that HOMER is the only method works reasonably well in the limit of 64K context length, it does seem to be an important work for the long-context learning of LLMs. Therefore, I decided to raise my score to 6.

---

> > > ### Author Response · Authors · 2023-11-23
> > > **Thank you for your response**
> > >
> > > Dear Reviewer X3So,
> > >
> > > Thank you for your comments and for taking the time to review our manuscript.
> > >
> > > We are happy that our additional discussion and experiments addressed your concerns!
> > >
> > > If you have any further comments or suggestions, please let us know. We are committed to improving the quality of our work, and
> > > we value your feedback.
> > >
> > > Thank you very much,
> > >
> > > Authors

---

> ### Author Response · Authors · 2023-11-16
> **Response to Reviewer X3So (2/3)**
>
> **[Q1] The inference time of HOMER was not discussed at all in the paper.**
>
> We appreciate your feedback that inference time is very important for LLM applications. In fact, **the inference speed of HOMER is significantly faster than the baselines**, thanks to the extensive computation reduction.
>
> The following table illustrates the average inference time for HOMER and other baselines. Specifically, we compare the time required to generate 20, 50, and 100 tokens, conditioned on 8k, 16k, and 32k contexts. We report a single value for all baselines following the setup in Section 4.5. The average inference time, calculated over 25 runs, is expressed in seconds. We also report the percentage of the speedup.
>
> | Tokens | Method      | 8k                    | 16k                   | 32k                    |
> |--------|-----------|-----------------------|-----------------------|------------------------|
> | 20     | Baselines | 1.879                 | 3.224                 | 6.546                  |
> |      | HOMER     | 1.673 (12.3% speedup) | 2.270 (42.0% speedup)    | 3.513 (86.3% speedup)  |
> | 50     | Baselines | 3.842                 | 6.028                 | 11.143                 |
> |      | HOMER     | 3.026 (27.0% speedup)   | 3.639 (65.6% speedup) | 4.873 (128.7% speedup) |
> | 100    | Baselines | 7.149                 | 10.733                | 18.828                 |
> |     | HOMER     | 5.355 (33.5% speedup) | 5.930 (81.0% speedup)    | 7.169 (162.6% speedup) |
>
> As evident from the results, HOMER provides a significant speedup (up to 162.6%) compared to the baseline methods. It's important to note that our method is even more effective while generating longer outputs conditioned on longer inputs, underscoring its effectiveness in handling long contexts.
>
> The main source of performance gain in HOMER is the computation reduction, elaborated in Section 4.5. The following points highlight these improvements:
> - The divide-and-conquer approach circumvents the quadratic computation associated with self-attention.
> - Token pruning significantly reduces the number of tokens to process, especially in the upper layers.
> - HOMER compresses long contexts into short embeddings, substantially reducing the size of the kv-cache. This lowers the computational demand during the decoding stage.
>
> We also emphasize that the computational overhead introduced by HOMER is minimal. The hierarchical context merging process involves relatively cheap operations, including matrix subtraction (calibration), gathering by index (calibration, token pruning and propagative refinement), top-k selection (token pruning), and tensor concatenation (merging). Conversely, it reduces more costly operations such as matrix multiplication for computing self-attention.
>
> We also added the above analysis in Appendix E.
>
> ---
> **[Q2-1] The explanation of propagative refinement is confusing. How are the refined embeddings gathered at the low-level layers?**
>
> For clarification, let us provide an alternative explanation of the process. In fact, propagative refinement is an additional refinement step taken after token reduction.
>
> The process is straightforward: when a token is pruned in the upper layers, the corresponding tokens (i.e. tokens that originated from the same input token) are also pruned in the lower-layer embeddings. This ‘propagates’ the pruning decision of the upper layers to the lower layers embeddings, hence the name. The synchronized pruning across layers results in shorter, uniform embeddings for each layer.
>
> For example, consider the situation where the tokens at positions 2, 3, and 5 are pruned in the N-th layer. By applying propagative refinement, we simply drop the tokens at positions 2, 3, and 5 from the lower-layer embedding, too.
>
> We also revised Section 3.2 with the above clarification, and added a step-by-step illustration of the process at Appendix D.
>
> ---
> **[Q2-2] How does the refined embeddings impact the computation results shown in the ablation Table 4(b)?**
>
> Although our motivation for propagative refinement was for computational efficiency, Table 4(b) shows that it is also beneficial for performance. This aligns with the recent findings in encoder-decoder models, where providing a large number of encoder outputs (that exceed the original context length) to the decoder often degrades performance, while selecting a few, relevant tokens is beneficial (see comparison of Unlimiformer and SLED in [9]). We believe that the superior performance of propagative refinement suggests that a similar phenomenon also happens for the decoder-only models.
>
> [9] Unlimiformer: Long-Range Transformers with Unlimited Length Input, NeurIPS 2023

---

> > ### Comment · Reviewer_X3So · 2023-11-22
> >
> > Thank you authors for the additional analysis of inference time comparison, and explanation on propagative refinement.

---

> ### Author Response · Authors · 2023-11-16
> **Response to Reviewer X3So (3/3)**
>
> **[Q3] How is K selected for top-K token pruning?**
>
> K was simply set to be the half of maximum chunk length, so that the resulting chunk after merging has the maximum length (in other words, we selected the largest K possible). For example in case of plain Llama + HOMER, K is 1024 because the maximum chunk length is 2048 (as mentioned in Appendix B). More extensive search on K could provide an even better balance between speed and performance.

---

> > ### Author Response · Authors · 2023-11-21
> > **A Gentle Reminder**
> >
> > Dear Reviewer X3So,
> >
> > Thank you very much again for your time and efforts in reviewing our paper.
> >
> > We kindly remind that we have only two days or so in the discussion period.
> > We just wonder whether there is any further concern and hope to have a chance to respond before the discussion phase ends.
> >
> > Many thanks,
> >
> > Authors

---

### Official Review · Reviewer_uJAL · 2023-10-31

**Soundness:** 3 good
**Presentation:** 3 good
**Contribution:** 3 good
**Rating:** 8
**Confidence:** 3

**Summary:**

This work presents a new approach to handle long context by LLMs without any training, by compressing and merging context chunks in a divide-and-conquer manner. Specifically, the long context is first split into chunks, and a LLM then encodes and compresses the chunk by pruning its tokens according to attention importance, finally concatenating adjacent chunks at each transformer layers. The proposed approach is shown achieving good performance on retrieval and QA tasks. Additionally, it can be viewed orthogonal to other techniques to extend LLM context length via scaling Rotary embeddings, and is shown state-of-the-art performance on QA when combing with other existing techniques together.

**Strengths:**

- The proposed approach brings a new perspective to handle long context by LLMs through context compression and merging, complementary to other popular techniques focusing on Rotary position embeddings.

- The approach is shown to surpass all baselines on Passkey Retrieval, extending to 8 times longer than the pretrained context length. For QA, it is shown to achieve the best performance combining with another long-context technique NTK.

- The proposed approach has the advantage of low GPU memory usage, compared with other baselines, due to the compression of context, leading to a reduced size of KV cache.

**Weaknesses:**

Though effective on the relatively easy Passkey Retrieval, the heavy compression and merging of context might hurt the reasoning capability of LLMs, as shown by Table 3 that the proposed approach alone underperforms other long-context techniques on QA. It is unknown whether this technique would remain useful on other tasks that also require reasoning. More experiments on other tasks could be conducted to demonstrate the advantages as well as the disadvantages of the proposed approach.

**Questions:**

What if we only look at the attention importance at the final layer directly, and use the top-k positions as the final pruning decisions for every layer? How is this compared to the proposed approach?

---

> ### Author Response · Authors · 2023-11-16
> **Response to Reviewer uJAL**
>
> Dear reviewer uJAL,
>
> We sincerely appreciate your efforts in reviewing our manuscript. We respond to each comment in the following content. We carefully incorporated the discussions into the revised manuscript. We highlighted the revised contents in blue for your convenience to check.
>
> ---
> **[W1] Heavy compression and merging may hurt the reasoning capability of LLMs, as shown in QA experiments. More experiments on other tasks could be conducted to evaluate this capability.**
>
> While it is true that the intensive compression might affect reasoning in certain scenarios, our evaluations in multiple-choice QA clearly demonstrate that the advantages of accessing extended contexts significantly outweigh this potential drawback. The ability of HOMER to handle longer contexts effectively enhances overall performance, improving the accuracy both when applied on the plain model and NTK. We have revised the layout of Tables 1, 2, and 3 to further emphasize this aspect.
>
> To further address the reasoning aspect, we conducted an additional evaluation using the Qasper benchmark, an open-ended QA framework [1]. The detailed experiment setup follows the QuALITY experiment, with F1 score as the performance metric. By applying HOMER, the plain model’s performance improved from 17.38 to 18.08 and YaRN’s performance improved from 19.08 to 20.33.
>
> These results not only affirm the efficacy of HOMER in contexts requiring reasoning but also highlight its compatibility and synergistic potential with other RoPE scaling methods.
>
> [1] A Dataset of Information-Seeking Questions and Answers Anchored in Research Papers, NAACL 2021
>
> ---
> **[Q1] What if we only look at the attention importance at the final layer directly, and use the top-k positions as the final pruning decisions for every layer? How is this compared to the proposed approach?**
>
> The suggested method will significantly increase the computational burden, as computing the significance scores in the final layer would require forwarding the full input across all layers. HOMER addressed this issue by iteratively pruning and merging chunks at the intermediate layers, resulting in the linear computation and logarithmic memory requirement.

---

> > ### Comment · Reviewer_uJAL · 2023-11-21
> > **Response Feedback**
> >
> > Thanks for the authors' response and adding the experiments on Qasper. I acknowledge that my concerns have been addressed.

---

> > > ### Author Response · Authors · 2023-11-21
> > > **Thank you for your response**
> > >
> > > Dear Reviewer uJAL,
> > >
> > > Thank you for your comments and for taking the time to review our manuscript.
> > >
> > > We are happy that our additional discussion and experiments on Qasper addressed your concerns!
> > >
> > > If you have any further comments or suggestions, please let us know. We are committed to improving the quality of our work, and we value your feedback.
> > >
> > > Thank you very much,
> > >
> > > Authors

---

### Author Response · Authors · 2023-11-16
**General Response**

Dear reviewers and AC,

We sincerely appreciate the time and effort you dedicated to reviewing our manuscript.

As highlighted by the reviewers, we believe our paper proposes a simple (ae6y) and effective (uJAL, X3So, wcXN) method that efficiently (uJAL, X3So) extends the context limit of LLMs, which can be easily adopted (ae6y) due to its training-free (uJAL, X3So, ae6y, wcXN) nature.

We appreciate your helpful suggestions on our manuscript. In accordance with your comments, we have carefully revised the manuscript with the following additional discussions and experiments:

- A clearer description of the method. (in Section 3.2, Section 3.3, Appendix D)
- Inference time analysis. (in Appendix E)
- A clearer presentation of the results. (in Table 1, Table 2, and Table 3)

We highlighted the revised contents in blue for your convenience to check. We sincerely believe that HOMER can be a useful addition to the ICLR community, especially, as the revision allows us to better deliver the effectiveness of our method.

Thank you very much!

Authors.

---

### Comment · Area_Chair_WxUk · 2023-11-20
**Please engage in reviewer-author discussions**

Reviewers - I encourage you to read the authors' response carefully and let the authors know whether their response has addressed your comments.

---

### Meta-Review · Area_Chair_WxUk · 2023-12-04

**Metareview:**

The paper introduces a training-free method to extend the context limit of LLMs (specifically 4K to 64K tokens using llama-2). The key idea is to divide the long context into manageable chunks, compress them by pruning tokens based on attention importance, and finally merge them hierarchically through the Transformer layers. This approach achieves promising performance on retrieval and QA against baseline methods, while avoiding the need of fine-tuning. During rebuttal, most of the reviewers have acknowledged that majority of the concerns in the initial reviews have been addressed. I comment on the major remaining concern about the language modeling experiment in the "Additional Comments On Reviewer Discussion" section below, as we haven't seen the reviewer's final response in the discussion thread. Overall, I think the paper has good clarity, novelty, and empirical results. Thus, I recommend accepting this work.

**Justification For Why Not Higher Score:**

For a spotlight, I would expect more breakthrough in some aspects, either theoretical or empirical, say beyond passkey retrieval and better results on QuALITY (the current results are ~3x%, which is only a bit better than random at 25%). I do understand the constraints on compute though.

**Justification For Why Not Lower Score:**

This work presents a novel approach with solid experiments and clear writing.

---

### Decision · Program_Chairs · 2024-01-16

Accept (poster)